# Optimized Route for the Fabrication of MnAlC Permanent Magnets by Arc Melting

**DOI:** 10.3390/molecules27238347

**Published:** 2022-11-30

**Authors:** Hugo Martínez-Sánchez, Juan David Gámez, José Luis Valenzuela, Hernan Dario Colorado, Lorena Marín, Luis Alfredo Rodríguez, Etienne Snoeck, Christophe Gatel, Ligia Edith Zamora, Germán Antonio Pérez Alcázar, Jesús Anselmo Tabares

**Affiliations:** 1Grupo de Metalurgia Física y Teoría de Transiciones de Fase, Departamento de Física, Universidad del Valle, A.A, Cali 25360, Colombia; 2Centro de Excelencia en Nuevos Materiales (CENM), Universidad del Valle, A.A, Cali 25360, Colombia; 3Departamento de Ciencias Básicas, Institución Universitaria António José Camacho, Avenida 6N No 28N-102 A.A, Cali 25663, Colombia; 4Grupo de Películas Delgadas, Departamento de Física, Universidad del Valle, A.A, Cali 25360, Colombia; 5Grupo de Transiciones de Fase y Materiales Funcionales, Departamento de Física, Universidad del Valle, A.A, Cali 25360, Colombia; 6CEMES-CNRS, 29 rue Jeanne Marvig, B.P. 94347, CEDEX 4, 31055 Toulouse, France

**Keywords:** rare-earth-free permanent magnets, electric arc melting, *τ*-MnAlC, *ε*-MnAlC

## Abstract

The rare-earth-free MnAlC alloy is currently considered a very promising candidate for permanent magnet applications due to its high anisotropy field and relatively high saturation magnetization and Curie temperature, besides being a low-cost material. In this work, we presented a simple fabrication route that allows for obtaining a magnetically enhanced bulk *τ*-MnAlC magnet. In the fabrication process, an electric arc-melting method was carried out to melt ingots of MnAlC alloys. A two-step solution treatment at 1200 °C and 1100 °C allowed us to synthesize a pure room-temperature *ε*-MnAlC ingot that completely transformed into *τ*-MnAlC alloy, free of secondary phases, after an annealing treatment at 550 °C for 30 min. The Rietveld refinements and magnetization measurements demonstrated that the quenched process produces a phase-segregated *ε*-MnAlC alloy that is formed by two types of *ε*-phases due to local fluctuation of the Mn. Room-temperature hysteresis loops showed that our improved *τ*-MnAlC alloy exhibited a remanent magnetization of 42 Am^2^/kg, a coercive field of 0.2 T and a maximum energy product, (BH)_max_, of 6.07 kJ/m^3^, which is higher than those reported in previous works using a similar preparation route. Experimental evidence demonstrated that the synthesis of a pure room-temperature *ε*-MnAlC played an important role in the suppression of undesirable phases that deteriorate the permanent magnet properties of the *τ*-MnAlC. Finally, magnetic images recorded by Lorentz microscopy allowed us to observe the microstructure and magnetic domain walls of the optimized *τ*-MnAlC. The presence of magnetic contrasts in all the observed grains allowed us to confirm the high-quality ferromagnetic behavior of the system.

## 1. Introduction

The face-centered tetragonal structure (*τ*-phase) of the near-equiatomic MnAl alloy, stabilized with a very low concentration of C, (commonly called *τ*-MnAlC) is currently considered very promising for permanent magnet applications due to its high anisotropy field (H_A_ ~ 4 T) and relatively high saturation magnetization (M_s_ = 96 Am^2^/kg) and Curie temperature (T_C_ = 380 °C) [1,2,3], besides being a low-cost material. Due to the need to develop rare-earth-free permanent magnets, from 2008, several works began to emerge that sought to establish the best procedure for obtaining the aforementioned *τ*-MnAlC phase with the best magnetic performance [4,5,6,7,8,9,10,11,12,13,14,15]. Typically, the *τ*-MnAlC phase is obtained from a quenched non-magnetic hcp ε-phase (ε-MnAlC), which is then annealed at temperatures between 500 and 700 °C [1,3,16,17] for a short time (15–40 min). Several studies conclude that the metastable nature of *τ*-phase results in its decomposition during prolonged annealing or high temperature processing [2,18,19,20]. Thus, throughout the fabrication process of the *τ*-MnAlC phase, some equilibrium phases such as β-Mn, Al-rich γ_2_ and/or Mn_3_AlC phases [2] can appear and, consequently, deteriorate the magnetic properties of the alloy.

Various techniques have been used to produce both MnAl- and MnAlC-based permanent magnets, such as arc-melting [13], induction melting [21], mechanical alloying [22], melt-spinning [8], mechanochemical synthesis [23] and gas atomization [24]. Jian et al. [13] used arc melting to produce *τ*-Mn_53.3_Al_45_C_1.7_, with a small amount of ε-phase, after heat treatment at 600 °C for 30 min, and reported magnetic properties of M_2T_ ~ 80 Am^2^/kg, M_r_ ~ 25 Am^2^/kg and $\mu_{0}$H_c_ ~ 0.1 T; Feng et al. [25] obtained (Mn_53_Al_45_C_2_)_98.5_Ni_1.5_ with a 91 wt.% of τ-phase by arc melting and reported a M_3T_ ~ 98 Am^2^/kg. Jiao et al. [26] reported a M_r_ = 1.44 kGs (23 Am^2^/kg, value estimated by assuming a theoretical density of 5.04 g/cm^3^ for the *τ*-MnAl [27]), a $\mu_{0}$H_c_ ~ 0.108 T and a (BH)_max_ ~ 2.0 kJ/m^3^ for (Mn_54_Al_46_)_98_C_2_ prepared by arc melting processes; Thongsamrit et al. [21] used induction melting to produce (Mn_55_Al_45_)_97_C_3_ with 50 wt.% of *τ*-phase after annealing at 500 °C for 20 min, and reported a M_1T_ ~ 21 Am^2^/kg; Shtender et al. [28] synthetized Mn_55_Al_45_C_2_ with 85 wt.% of *τ*-phase and obtained a M_8.5T_ = 541 kA/m (108 Am^2^/kg) and a $\mu_{0}$H_c_ = 0.0715 T. Recently (2021), Feng et al. [29] produced Mn_53_Al_45_C_2_ by induction melting with 100 wt.% of *τ*-phase after subsequence annealing for 48 h at 1100 °C, followed by 500 °C for 1 h, and reported a M_5.6T_ ~ 0.75 T (119 Am^2^/kg); Saito et al. [22] produced Mn-Al-C (70 wt.% Mn, 29.5 wt.% Al, 0.5 wt.% C) through mechanical alloying with about 40 wt.% of *τ*-phase, and after annealing at 1100 °C for 1 h, they reported M_1.5T_ ~ 40 Am^2^/kg, Mr ~ 24 Am^2^/kg and $\mu_{0}$H_c_ = 0.23 T; Palmero et al. [24] implemented gas atomization to produce Mn-Al-C alloys with a remanent of equilibrium γ_2_ phase, and they reported M_2T_ = 65 Am^2^/kg and $\mu_{0}$H_c_ = 0.163 T. As we have seen above, the synthesis of a *τ*-MnAlC permanent magnet in bulk, with no secondary phases and good magnetic properties, remains a challenge for researchers; producing a high-quality *τ*-phase is very important as it opens the way to further improving the performance of micro- or nanostructured *τ*-MnAlC magnets. 

Some of the most attractive routes to optimize the magnetic properties of a *τ*-MnAlC magnet are mechanical milling [4]; hot extrusion [30]; hot pressing [31]; mixtures with other elements such as Mn_65_Ga_35_ [26]; exchange springing with soft magnetic phases such as α–Fe [32], Fe_65_Co_35_ [33,34] and MnB [35]; and thin film processes [36]; and each of them has advantages and disadvantages. For instance, the hot extrusion method allows the production of highly anisotropic MnAl-C-Ni magnets with a high magnetic energy (BH)_max_ = 43 kJ/m^3^ [30], but it uses an expensive process that is not suitable for large-scale production; mechanical milling is a cost-effective and efficient process to improve the magnetic performance of MnAl alloys with nanocrystalline microstructures [37,38], and is beneficial for increasing the coercivity ($\mu_{0}$H_c_ = 0.5 T [38]), but it produces a large amount of equilibrium phases after the thermal treatment following the milling process, leading to reduced magnetization [4,39]. Due to the relevance of producing a magnetically enhanced bulk *τ*-MnAlC permanent magnet, the main goal of this work is to optimize a route to produce a high-quality *τ*-MnAlC magnet by electric arc melting and validate the reproducibility of the process.

## 2. Materials and Methods

### 2.1. Preparation of the As-Cast MnAlC

Alloy ingots with nominal composition Mn_53.3_Al_45_C_1.7_ (refered here as MnAlC) were prepared by arc melting using high-purity (>99.9%) powders of Mn, Al and C, under a high-purity argon atmosphere, and melted four times in order to ensure the formation of a homogeneous mixture. As a first step, Al and C powders were manually mixed and introduced into a steel die to compact it under a pressure of 10.3 MPa. This compaction procedure is necessary to avoid C loss during the melting process and to obtain a better diffusion of the carbon within the Mn-Al matrix. In parallel, powder of Mn was compacted under a pressure of 20.6 MPa, and subsequently introduced into an electric arc furnace in a small cup machined on a cooled cooper plate that forms the base of the arc furnace chamber. Prior to arc melting, a purge chamber procedure is performed, in three stages, for 30 min. The current intensity used was close to 65 A. Under these conditions, an electrical arc is formed to cover the samples. The compacted Mn was molten twice. When obtaining the molten Mn, in a second melting process, a compacted Al-C system was added in the desired stoichiometry. The molten Mn and compacted Al-C systems were placed in the mold of the cooled cooper plate, with the compacted Al-C sample placed on top, the molten Mn in the bottom and the tungsten electrode, at 2 cm apart. This was intentionally decided so the arc did not interact directly with the molten Mn at the beginning of melting process. In addition, an extra amount of Mn (3.0 wt.%) was added to compensate the Mn that evaporates during the melting processes [32,33].

### 2.2. Preparation of the Quenched ε–MnAlC

For the preparation of the room-temperature (RT) *ε*-MnAlC alloy, the as-cast MnAlC ingots were encapsulated into a quartz tube under an argon atmosphere, at a pressure of 0.6 MPa, and then two different solution treatment routes were performed: (Route I), one as-cast MnAlC ingot (referred to here as Sample-A) was heated from room temperature to 1100 °C, with a heating rate of 2 °C/min, keeping this temperature constant for 32 h, and finally quenched the sample in RT water; (Route II), a second as-cast MnAlC sample (referred to here as Sample-B) was directly introduced to the furnace after it reached a temperature of 1200 °C, such a temperature was maintained for 4 h, and then it was reduced to 1100 °C and kept constant for 48 h, before the sample was finally quenched in RT water. The schemes of the above-mentioned solution treatment routes are shown in Figure 1a,b, respectively. Route I is inspired by reference [13], and Route II is a modified Route I that, as we will show later, allows for improving the fabrication of high-quality *τ*-MnAlC magnets.

### 2.3. Annealing Treatment to Obtain the τ-MnAlC

The thermal procedure to transform the RT *ε*-MnAlC to the *τ*-MnAlC phase takes into account two fundamental parameters: the annealing temperature and time [4,8,12,13]. In our study, we set the annealing temperature at 550 °C, according to a previous study [33] and varied the annealing time (15, 20 and 30 min) in order to find the suitable condition to produce a virtually pure *τ*-MnAlC alloy. A scheme of the annealing process is shown in Figure 2a. To perform the isothermal annealing, the quenched *ε*-MnAlC alloys were introduced into a quartz tube under an argon atmosphere, at a pressure of 0.6 MPa, and put them inside the furnace when it reached 550 °C. After finishing the annealing time, the resulting MnAlC alloys were quenched in RT water.

### 2.4. Characterization Methods

The crystal structure and composition of the alloys was examined by X-ray diffraction (XRD) measurements carried out in a X’Pert PRO MRD diffractometer (PANalytical, Malvern, UK), using the Bragg–Brentano configuration with a step size of 0.0197°, a step time of 141.740 s and a Cu Kα radiation (wavelength of 1.5406 Å). For all diffractograms, a Rietveld refinement was carried out to detect the different crystal phases and quantify their concentrations in the resulting alloys. A magnetic characterization of the samples was performed through room-temperature magnetization hysteresis loops measured in a PPMS (Quantum Design, San Diego, CA, USA), using the vibrating sample magnetometer (VSM) mode. Hysteresis loops were measured using 1.8 T and 3 T as the maximum applied fields. A microscopy analysis of grain size and magnetic microstructure of an optimized *τ*-MnAlC alloy was carried out in a Hitachi HF-3300 (I2TEM-Toulouse) microscope, operated at 300 kV and configurated to perform Lorentz microscopy, a technique capable of imaging magnetic domain walls.

## 3. Results and Discussion 

### 3.1. XRD Characterization

Figure 1 displays the two solution treatment routes followed in this work to obtain the RT *ε*-MnAlC samples and, below them, their respective XRD patterns. According to the XRD analysis of Figure 1c, Route I produces a heterogeneous mixture (referred to here as *ε*-Sample-A) made up of *ε*-MnAlC, *τ*-MnAlC and Al_6_Mn phases. Inspired by previous works [40,41], the best refinement was achieved by considering two types of *ε*-phases (referred to here as *ε*_1_ and *ε*_2_) where the Mn content is slightly varied, taking into account the upper and lower limits within which this phase is stable, according to the Mn-Al phase diagram [42]. Here, we define that *ε*_1_ is formed by 42.5 at.% of Al and 57.5 at.% of Mn (a Mn-rich *ε*-phase) and *ε*_2_ is formed by 47.5 at.% of Al and 52.5 at.% of Mn (a Mn-poor *ε*-phase). As shown in Table 1, Route I synthesized a quenched MnAlC alloy that is composed of a high-concentration *ε*-phase (~86 wt.%), in which *ε*_2_ is the majority phase, with small quantities of *τ*-MnAlC and Al_6_Mn. The presence of an Al-rich phase evidences that the solution treatment followed in Route I does not favor the complete homogeneous distribution of the constituent elements along the sample to form a unique high-temperature *ε*-MnAlC phase; the formation of the *τ*-MnAlC during the quenching process has been reported previously [13], and it could be associated with incomplete atomic diffusion by thermal stresses, induced in the quenching process due to the thermal gradient [43].

It is well known that the formation of additional phases in quenched *ε*-MnAlC-based alloys prevents the synthesis of a high-quality *τ*-MnAlC after the annealing treatment [13,21,25]. For this reason, we propose a modified Route I (here referred to as Route II) where a two-step solution treatment favors the homogenization of the sample. According to the XRD analysis of Figure 1d, Route II allowed the quenching of a pure *ε*-MnAlC, virtually free of secondary phases. Similar to Route I, a better Rietveld refinement was obtained considering a two-phase segregation of *ε*-MnAlC, where *ε*_1_ was the dominant phase. Therefore, we observe that this modified solution treatment, which provides sufficient energy to the constituent elements of the system, can randomly distribute along the sample and consequently improve the homogenization of the resulting *ε*-MnAlC alloy.

Figure 2b displays the XRD patterns of the annealed *ε*-MnAlC alloys for different annealing times (15, 20 and 30 min) at 550 °C. This systematic study was performed on the pure quenched *ε*-MnAlC ingots (*ε*-Sample-B) and allowed us to determine that an annealing time of 30 min is enough to produce a pure *τ*-MnAlC alloy (*τ*-Sample-B, Figure 2d. Using a lower annealing time, the *ε*-MnAlC can also decompose in the equilibrium β-Mn phase (something we expected). The optimized annealing process to produce the *τ*-MnAlC was also performed in the heterogeneous mixture (*ε*-Sample-A). An XRD pattern of this annealed sample (here referred to as *τ*-Sample-A) is displayed in 2 (c) and shown as the annealing treatment, allowed for increasing the amount of the *τ*-MnAlC initially present in the quenched sample, due to the *ε*→*τ* transformation process. The concentration of Al_6_Mn also increased, but not significantly, so that it does not contribute much to the deterioration of the desired *τ*-phase during the annealing process. However, the annealing treatment also favored the formation of the β-Mn that comes from the decomposition of *ε*-MnAlC phases. Thus, we demonstrate that an inhomogeneous as-cast MnAlC-based alloy prevents the obtaining of a pure *τ*-MnAlC alloy.

In order to verify the reproducibility of the method, an additional sample was produced (referred to here as Sample-C), following steps 1 and 2 (Figure 3a,b) to synthesize a pure *τ*-MnAlC alloy. The Rietveld refinements performed in the XRD patterns taken in both *ε*-Sample-C and *τ*-Sample-C systems show that the two-step solution treatment (Route II) again favored the production of a pure RT *ε*-MnAlC alloy that completely transformed into a *τ*-MnAlC alloy, free of secondary phases. Similar to *ε*-Sample-A and *ε*-Sample-B, the XRD pattern of *ε*-Sample-C was refined by assuming a two-phase segregation of the *ε*-MnAlC, where *ε*_2_ was the dominant phase, a result similar to *ε*-Sample-A. In terms of crystalline structure and composition, we detected no significant difference between *τ*-Sample-B and *τ*-Sample-C, and both have a pure *τ*-MnAlC phase.

### 3.2. Magnetic Characterization

Figure 4 shows the magnetization curves for all synthesized samples, and Table 2 summarizes the main magnetic parameters. We find that all *ε*-MnAlC-based alloys exhibited a small hysteretic behavior, similar to that observed in ferromagnetic materials (see Figure 4a). In the case of *ε*-Sample-A, a ferromagnetic response was expected because of the presence of the *τ*-phase. However, for *ε*-Sample-B and *ε*-Sample-C, their ferromagnetic behavior should come from the Mn-poor *ε*-phase regions (*ε*_2_), which should have a ferromagnetic character [41]. Thus, these magnetization measurements proved the two-phase segregation of *ε*-MnAlC, where local fluctuations induce a phase separation into antiferromagnetic (Mn-rich) and ferromagnetic (Mn-poor) regions.

In the *τ*-MnAlC-based systems, the ferromagnetic behavior is only due to the *τ*-phase. As could be expected from the XRD analysis, *τ*-Sample-A showed a low remanent magnetization because only 60% of the total mass of the sample is ferromagnetic. Furthermore, it also showed a reduced coercive field, even lower than those observed in the hysteresis loops of the *ε*-phase-based alloys (see Table 2), so the presence of secondary phases such as β-Mn and Al_6_Mn does not promote magnetic coupling and/or pinning effects that would favor the coercivity of the alloy. Although they were synthesized using the same route, the magnetization curves of *τ*-Sample-B and *τ*-Sample-C showed some differences: the magnetization values in the first and third quadrants are lower for *τ*-Sample-B (e.g., there is a percentage difference of around 23% at 1.8 T). However, throughout the magnetization reversal process, which takes place in the second (or fourth) quadrant, the two curves seem to be similar, reaching the same coercive field (*μ*_0_H_c_ = 0.2 T) and with remanent magnetizations with a percentage difference close to 10%. Considering that both systems only have the *τ*-phase, we assume that the differences observed in the magnetization curves should be related to the resulting crystalline microstructure, which would not be the same for both; it is affected by dislocations, twin structures and stacking faults that emerge during the *ε*→*τ* transformation [44,45,46,47,48], which we cannot control. Similar hysteresis loop differences were experimentally observed by Jia et al. [47] in deformed *τ*-MnAlC samples where an increase in the dislocation density caused a substantial reduction of the magnetization, impacting H_c_ and M_r_ slightly; and theoretically observed [48] by simulating twin-free and twinned coarse grains/particles. In this work, a detailed explanation of the origin of the magnetic differences between *τ*-Sample-B and *τ*-Sample-C is beyond the scope of this study, and an exhaustive characterization that combines micro-, macro- and in-situ characterization is required. However, it is important to highlight that both samples presented close values of the maximum energy product (~5% difference), with a value of 6.07 kJ/m^3^ for *τ*-Sample-C, the alloy with the best magnetic properties for this study. The values of M_r_ and *μ*_0_H_c_ for *τ*-Sample-B and *τ*-Sample-C are better than those reported in as-transformed *τ*-MnAl (C)-based alloys synthesized by the arc-melting method [12,13,25,26], and only the use of hot-deformation and hot-extrusion processing can produce samples with better properties than those obtained here [49].

Finally, Figure 4 shows Thamm–Hesse curves obtained from de hysteresis loops to indirectly identify the magnetic interactions between the clusters/grain/defects forming the samples. Here, the Δ*M* curve is reconstructed by the equation, $\Delta M=M_{ic}-1/2\;\left(M_{uc}+M_{lc}\right)$, where *M*_*ic*_ corresponds to the initial magnetization curve, and *M_uc_* and *M_lc_* are associated with the upper and lower branch of the hysteresis loop, respectively [50]. In the *τ*-MnAlC-based alloys, all the Δ*M* curves were deviated to negative values, as expected for FM *τ*-phases [4,32], evidencing a dominant demagnetization effect due to dipolar interactions between single-domain powder particles, local antiferromagnetic coupling between crystalline defects such as the stacking faults [51], and coupling between a magnetically disordered surface and an ordered core, in the case of nanometer grains [52,53]. Similar behavior was observed in most of the *ε*-MnAlC-based alloys, except *ε*-Sample-A, where a positive deviation was observed at fields higher than 1.8 T, showing that its compositional inhomogeneities can induce exchange interactions that are dominant at high fields.

### 3.3. TEM Characterization

An as-transformed ingot of *τ*-Sample-C was selected for nanoscale observation by TEM, using a special mode that can resolve magnetic contrasts of ferromagnetic materials: Lorentz microscopy (LM). LM images recorded in an electropolished TEM sample are shown in Figure 5. The intermediate-magnification TEM image of Figure 5a revealed that our synthesis process produces a polycrystalline sample consisting of micrometer grains of different sizes and irregular shapes. Just in this image, we could identify grains of hundreds of nanometers of lateral size and very large grains of tens of microns. In some grains, we observed stacking faults and crystal defects that were created during the *ε*→*τ* transformation [44,47]. In addition, we did not detect the presence of micro- or nanosize grains of the *ε*-MnAlC or any other secondary phase.

Figure 5b shows a LM image taken in a region of the sample. In this image, the bright and dark contrasts are produced by magnetic domain walls formed inside of the grains. As we can see, domain walls are formed inside all the grains, and therefore we could see that most grains (if not all) correspond to the *τ*-MnAlC. More Lorentz images were taken in other regions of the sample and all the grains showed magnetic contrast; this means that, presently, we could say that Route II allows us to produce a high quality *τ*-MnAlC phase.

## 4. Conclusions

We presented an optimized manufacturing route that produced magnetically enhanced *τ*-MnAlC magnets using arc melting, a simple and inexpensive technique. The use of a two-step solution treatment at 1200 °C (for 4 h) followed by 1100 °C (for 48 h) has enabled us to synthesize pure *ε*-MnAlC alloys that presented a two-phase segregation condition. The high-purity condition of the RT *ε*-MnAlC alloys made it possible to produce pure *τ*-MnAlC alloys by annealing at 550 °C for 30 min, with permanent magnet properties (remanent magnetization and coercive field) that exceed most of the values previously reported in magnetic MnAlC alloys prepared by melting processing. As observed in the *τ*-MnAlC alloy prepared by a conventional route (Route I), the presence of secondary phases (even in small quantities) in the RT *ε*-MnAlC strongly affects the purity of the *τ*-MnAlC alloys and significantly deteriorates their permanent magnet properties. 

The XRD analysis showed that the two *τ*-MnAlC alloys prepared by the same route are similar, validating the reproducibility of the method, in terms of crystal composition and structure. Although the magnetization hysteresis loops showed substantial differences between the two samples, they had similar values for coercive field and magnetization remanence, so their maximum energy product of approximately 6 kJ/m^3^ was comparable. This can be improved through a mechanical process of grain-size reduction that increases coercivity, and/or by mixing it with a hard magnetic material which, by exchange spring coupling, could increase its remanence.

Finally, a local exploration of the microstructure of the improved *τ*-MnAlC alloy showed that our fabrication process produces a polycrystalline sample formed by micrometer grains of different sizes and irregular shapes, and some of them presented stacking faults and crystal defects created during the *ε* to *τ*-MnAlC transformation.

## Figures and Tables

**Figure 1 molecules-27-08347-f001:**
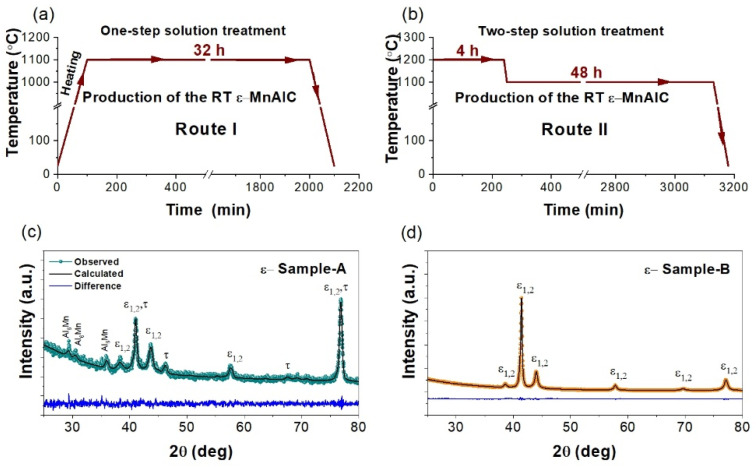
Schematic representation of the solution treatment routes: (**a**) Route I and (**b**) Route II. XRD pattern taken on the resulting samples from (**c**) Route I (*ε*-Sample-A) and (**d**) Route II (*ε*-Sample-B).

**Figure 2 molecules-27-08347-f002:**
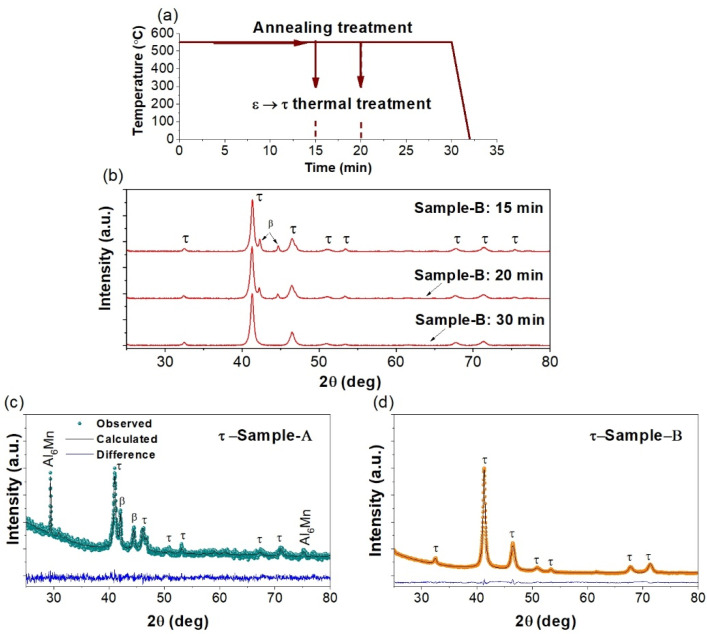
(**a**) Schematic graph of the annealing process to produce the *τ*-MnAlC at different annealing time. (**b**) XRD patterns of the resulting phase alloys after annealing the pure *ε*-MnAlC (*ε*-Sample-B) at different annealing times. (**c**,**d**) are XRD patterns for the resulting alloys after annealing *ε*-Sample-A and *ε*-Sample-B samples, respectively.

**Figure 3 molecules-27-08347-f003:**
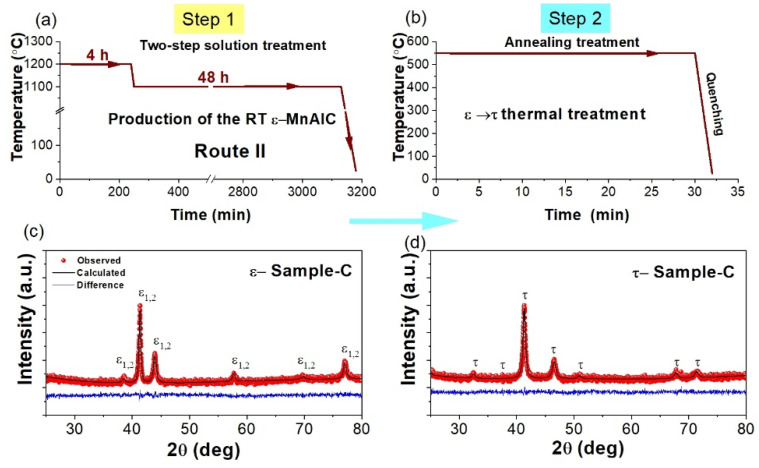
Schematic representation of the thermal treatments (Route II) followed to produce *τ*-phase from (**a**) step 1 and (**b**) step 2. (**c**,**d**) are XRD patterns taken in the resulting sample from (**a**,**b**), respectively.

**Figure 4 molecules-27-08347-f004:**
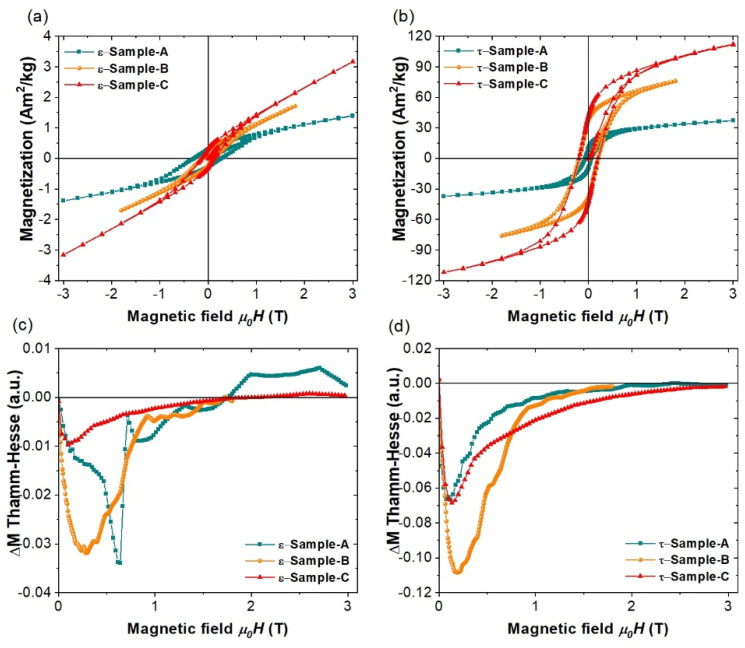
Magnetic hysteresis loops for all samples in the (**a**) quenched [*ε*] and (**b**) annealed [*τ*] states. (**c**,**d**) are Thamm–Hesse curves estimated from (**a**,**b**).

**Figure 5 molecules-27-08347-f005:**
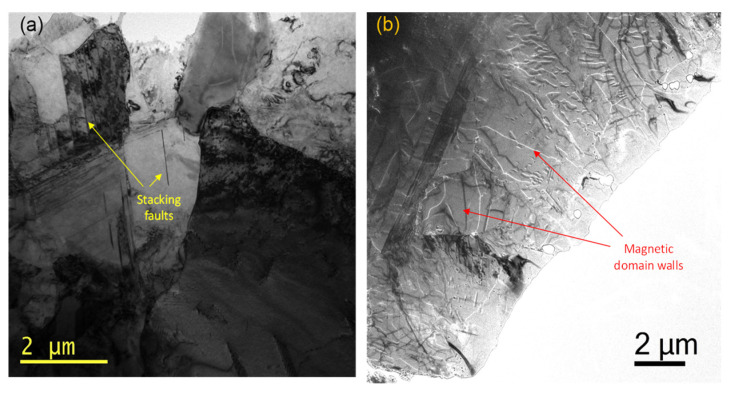
TEM characterization for *τ*-Sample-C (**a**) TEM image at intermediate magnification and (**b**) magnetic image by Lorentz microscopy.

**Table 1 molecules-27-08347-t001:** Summary of the main parameters obtained from the Rietveld refinements for samples *ε*-Sample-A, *ε*-Sample-B and *ε*-Sample-C, and their resulting alloys after the annealing treatment (*τ*-Sample-A, *τ*-Sample-B and *τ*-Sample-C).

Sample	Phases	Lattice Parameter (Å)	Weigth Fraction (wt.%)	Estimated Density (g/cm^3^)	χ^2^	R (F^2^)
*ε-Sample-A*	*ε* _1_	*a* = 2.705 *c* = 4.405	5	5.12	0.9264	0.1165
	*ε* _2_	*a* = 2.700 *c* = 4.383	81	5.00		
	*τ*-MnAlC	*a* = 3.914 *c* = 3.604	10	5.11		
	Al_6_Mn	*a* = 7.798 b = 6.473 *c* = 8.801	4	3.24		
*ε-Sample-B*	*ε* _1_	*a* = 2.704 *c* = 4.376	69	5.16	1.4330	0.0170
	*ε* _2_	*a* = 2.706 *c* = 4.381	31	4.98		
*ε-Sample-C*	*ε* _1_	*a* = 2.697 *c* = 4.370	11	5.19	0.7139	0.0542
	*ε* _2_	*a* = 2.670 *c* = 4.379	89	5.01		
*τ-Sample-A*	*τ*-MnAlC	*a* = 3.935 *c* = 3.590	60	5.08	0.0863	0.6470
	β-Mn	*a* = 6.440	33	6.83		
	Al_6_Mn	*a* = 7.524 *b* = 6.459 *c* = 8.914	7	3.32		
*τ-Sample-B*	*τ*-MnAlC	*a* = 3.919 *c* = 3.596	100	4.93	3.4690	0.0420
*τ-Sample-C*	*τ*-MnAlC	*a* = 3.919 *c* = 3.595	100	5.11	0.7497	0.0798

**Table 2 molecules-27-08347-t002:** Summary of the magnetic parameters.

	*ε*-Phase			*τ*-Phase			
	M_1.8T_ (Am^2^/kg)	*µ*_0_H_c_ (T)	M_r_ (Am^2^/kg)	M_1.8T_ (Am^2^/kg)	*µ*_0_H_c_ (T)	M_r_ (Am^2^/kg)	(BH)_max_ (kJ/m^3^)
Sample-A	1.03	0.29	0.29	32.81	0.068	9.57	------
Sample-B	1.71	0.12	0.25	76.04	0.20	37.62	5.79
Sample-C	2.14	0.13	0.30	98.6	0.20	42.12	6.07

## Data Availability

The date that support the results of this research are available when the editorial office of this journal requires it.

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
