# Peer review of "Optimized Route for the Fabrication of MnAlC Permanent Magnets by Arc Melting"

_molecules, 2022, doi:10.3390/molecules27238347_

Round 1

Reviewer 1 Report

General comments

This work investigates the fabrication of MnAlC alloy for its application as a permanent magnet, using an optimized process. First, ingots are prepared by arc-melting, then they are annihilated with different protocols and rapidly quenched at room temperature in water, to obtain the epsilon-MnAlC alloy. Finally, the conversion from epsilon to tau-MnAlC is obtained with additional mild annealing. The MnAlC alloy is a promising rare-hearth-free candidate to produce highly efficient permanent magnets. In this framework, the goal is to produce the FM tau-phase with the best magnetic properties, i.e., large coercive field and saturation magnetization, to produce the largest energy product possible. This work introduces a two-step first annealing using slow heating up to 1200 for a relatively short time (4h) to improve the homogeneity of the elements, followed by a long step (48h) at 1100°c, before a final rapid quenching to room temperature in water. The results show that an intermediate alloy with only the epsilon-phase fully converts into a tau-phase with improved magnetic properties with the following annealing.

The production of efficient permanent magnets with low environmental impact is of extreme interest, and the results presented in this work are promising and offer a clear improvement of the final MnAlC product. The solid structural characterization corroborates the discussed results, however, the magnetic study of the samples required further clarification, to let the quality of this work fully emerge.

Detailed comments

The detailed Rietveld analysis is distributed in two tables. However, the analysis of the eplison-phase in all samples proposed in table 1 is corrected later with a more refined model of the sample and summarized in table 2. I would suggest considering presenting a single table, with only the final selected multiphase model used to reproduce the patterns of the epsilon-phases. Moreover, table 1 presents the results for sample C, which is shown only later, but there is no mention of it in the caption of the table.

The magnetic characterization analyzed the epsilon and tau-phases showing a clear improvement for the novel annealing. However, the estimation of a low fraction of tau-phase in the intermediate epsilon alloy does not seem to agree with the results of the Rietveld refinement. The maximum magnetization at a high field is influenced not only by the ferromagnetic but also by the paramagnetic fraction of the sample. A more reasonable comparison can be made on the base of the remanent magnetization, being this an intrinsic characteristic of the FM phase only. If we compare Mr for the three samples in the epsilon form, the values are very close. It is difficult to believe that sample A contains 29% of the FM tau-phase as defined by the Rietveld analysis, while e this amount is only 0.6% for the other two samples. The authors should clarify this result.

The M(H) curves for sample B are measured only up to 1.8 T compared to 3 T used for samples A and C. What is the reason for this? Since the samples present a mixture of different phases and considering possible magnetic frustration due to the simultaneous presence of ferromagnetic (FM) and antiferromagnetic (AFM) exchange coupling, there is a strong possibility of recording minor loops and/or tilting of the canted structure at the highest field producing irreversible effects, once more, responsible of different remanence results after different saturation fields. In this framework, measuring the samples in the same condition is mandatory to be able to compare the results in a meaningful way.

The use of the Tham-Hesse curve is an interesting alternative to the more conventional Henkel plots to estimate the magnetic interactions among clusters/grains forming a magnetic sample. In both types of curves, negative deviations are the signature of demagnetization effects among the particles, while positive ones indicate spontaneous parallel alignment in an FM-like fashion of the magnetization of the elements. In the case of interacting single-domain particles, the demagnetization effect is typically induced by dipolar interactions. O the other hand, the positive signal due to FM-like alignment can be ascribed to FM-coupling through direct exchange involving atoms at the particle interfaces. Now, considering the size of the microsized grains forming the samples, the picture is more complex, since this involves multidomain grains and the formation/displacement of domain walls during the magnetization reversal. Hence, the deviations in the curves can have multiple origins and this should be commented on in the text. In general, figure 4c, and even more 4d, presents magnetic samples with dominant demagnetization effects that can be due to dipolar interactions, but one cannot neglect the presence of AFM atomic exchange inside each grain which is expected to dominate over magnetostatic interactions in a similar scenario with very large grains. Considering the expected mixed site occupancy leading to local AFM coupling, often localized also at crystalline defects, like the stacking faults evidenced in figure 5a, this seems a plausible scenario. Recently, frustrated AFM structures in small nanoparticles have been shown to induce strong dominant negative deviations in Henkel plots besides the intensity of the dipolar interactions.

https://doi.org/10.1186/s11671-022-03737-w

https://doi.org/10.1021/acs.chemmater.7b02522

Author Response

The Editor

Molecules Journal of the Multidisciplinary Digital Publishing Institute (MDPI)

November 7th, 2022

Dear Editor,

Thank you for considering our manuscript with ID: molecules-1984915 for publication in Molecules.  I am grateful to you and the reviewers for the valuable suggestions provided. They have all been considered and we have revised our manuscript following their suggestions.

Here are responses to the reviewer #1 comments:

Dear reviewers, we are very grateful for all your comments on our manuscript, which have led us to rethink some analyses, looking for the best presentation of the results that enables us to improve the quality of the manuscript. Major corrections were done, they were introduced in the manuscript using the “Track Changes” tool for Word.

  • The detailed Rietveld analysis is distributed in two tables. However, the analysis of the eplison-phase in all samples proposed in table 1 is corrected later with a more refined model of the sample and summarized in table 2. I would suggest considering presenting a single table, with only the final selected multiphase model used to reproduce the patterns of the epsilon-phases. Moreover, table 1 presents the results for sample C, which is shown only later, but there is no mention of it in the caption of the table.

R// We agree to present only one table (Table 1) with the model of the two epsilon phases. We have included the following sentence (lines 167-171): “Inspired by previous works [40,41], the best refinement was achieved by considering two types of e-phases (referred here as e1 and e2) where the Mn content is slightly varied, taking into account the upper and lower limits within which this phase is stable, according to the Mn-Al phase diagram [42]”. In addition, we have redone the Rietveld refinement of the e-Sample-A, assuming the existence of the two types of e-phases; this new refinement allowed to improve the c2 and substantially reduce the weight fraction of the t and Al6Mn phases, a result that is more coherent with the magnetic measurements. This correction motivated to make important changes in the discussion of the results.  

On the other hand, we have corrected the caption of Table 1 to mention e-Sample-C and t-Sample-C.

  • The magnetic characterization analyzed the epsilon and tau-phases showing a clear improvement for the novel annealing. However, the estimation of a low fraction of tau-phase in the intermediate epsilon alloy does not seem to agree with the results of the Rietveld refinement. The maximum magnetization at a high field is influenced not only by the ferromagnetic but also by the paramagnetic fraction of the sample. A more reasonable comparison can be made on the base of the remanent magnetization, being this an intrinsic characteristic of the FM phase only. If we compare Mr for the three samples in the epsilon form, the values are very close. It is difficult to believe that sample A contains 29% of the FM tau-phase as defined by the Rietveld analysis, while e this amount is only 0.6% for the other two samples. The authors should clarify this result.

R// We fully agree with the referee’s claim regarding that the magnetization value at a certain magnetic field is not a suitable parameter for estimating the ferromagnetic fraction in each system; and even more after doing another review of the literature. For this reason, we have removed these estimations from Table 2 (Table 3 in the previous version of the manuscript).

However, we have found that the e ® t transformation could produce dislocations, twin structures and stacking faults that strongly impact the ferromagnetic response of the t-MnAlC. For instance, the formation of a twin structure reduces both remanence and coercivity (see Phys. Rev. Materials 4, 094402 (2020); Phys. Rev. Materials 5, 064403 (2021)) due to different factors such as they promote the nucleation process of magnetic domain walls and do not enhance the pinning field (it reduces the coercivity) and they contain excess Mn atoms that reduce the local magnetization (it reduces the remanence). In the case of dislocations, they strongly affect how the magnetization of the system increases with the applied magnetic field (for a particular magnetic field, the higher the dislocation density, the lower the magnetization); however, the variation of Mr and Hc is less dramatic, so small changes are observed when there is not a great variation in the dislocation density. The latter could explain the discrepancies between t-Sample-B and t-Sample-C.          

Therefore, we have introduced the following changes in the manuscript: (1) We have removed the column headed %t; (2) we have rewritten the section 3.2 “Magnetic characterization” in order to be more consistent with these new findings.

  • The M(H) curves for sample B are measured only up to 1.8 T compared to 3 T used for samples A and C. What is the reason for this? Since the samples present a mixture of different phases and considering possible magnetic frustration due to the simultaneous presence of ferromagnetic (FM) and antiferromagnetic (AFM) exchange coupling, there is a strong possibility of recording minor loops and/or tilting of the canted structure at the highest field producing irreversible effects, once more, responsible of different remanence results after different saturation fields. In this framework, measuring the samples in the same condition is mandatory to be able to compare the results in a meaningful way.

R// According to Thamm-Hesse plots (Figure 4), at 1.8 T most of the sample present a DM which is very close to 0, or negative, meaning that at such field the initial, descending, and ascending magnetization curves coincide, so we are in the reversible regime (ahysteretic part) of the magnetization loop. Therefore, the hysteresis loop of t-Sample-B corresponds to the major one. We decide to use higher fields to Samples A and C to visualize the positive value of DM for e-Sample-A, and because the ahysteretic part of t-Sample-C begins close to 3 T. Although all the magnetic phenomena described by the referee may be present in t-phase based alloys, when secondary phase are formed, we believe the difference in magnetization between Sample B and C is than that discussed in the previous comment.

  • The use of the Tham-Hesse curve is an interesting alternative to the more conventional Henkel plots to estimate the magnetic interactions among clusters/grains forming a magnetic sample. In both types of curves, negative deviations are the signature of demagnetization effects among the particles, while positive ones indicate spontaneous parallel alignment in an FM-like fashion of the magnetization of the elements. In the case of interacting single-domain particles, the demagnetization effect is typically induced by dipolar interactions. O the other hand, the positive signal due to FM-like alignment can be ascribed to FM-coupling through direct exchange involving atoms at the particle interfaces. Now, considering the size of the microsized grains forming the samples, the picture is more complex, since this involves multidomain grains and the formation/displacement of domain walls during the magnetization reversal. Hence, the deviations in the curves can have multiple origins and this should be commented on in the text. In general, figure 4c, and even more 4d, presents magnetic samples with dominant demagnetization effects that can be due to dipolar interactions, but one cannot neglect the presence of AFM atomic exchange inside each grain which is expected to dominate over magnetostatic interactions in a similar scenario with very large grains. Considering the expected mixed site occupancy leading to local AFM coupling, often localized also at crystalline defects, like the stacking faults evidenced in figure 5a, this seems a plausible scenario. Recently, frustrated AFM structures in small nanoparticles have been shown to induce strong dominant negative deviations in Henkel plots besides the intensity of the dipolar interactions.

R// We thank the referee for helping us to improve the analysis regarding the Thamm-Hesse curves. We have included your comments in the manuscript.

Reviewer 2 Report

Manuscript is well organised and conclusions are clear, but the points mentioned below which should be addressed in a revised version.

The authors present a modified preparation route for the fabrication of MnAlC permanent magnets based on arc melting. Aim is to produce in a first step as pure as possible ε-AlMnC phase, which in a second step is transformed to the magnetic τ-phase. Three samples are discussed: Samples A, B, and C. Whereas sample A is prepared according to ref. 13 (route I), samples B and C are prepared in a slightly different new way (route II). Phases and magnetic properties are investigated by XRD and dc magnetic measurements. It is clearly shown, that Route II leads to more homogenous samples with better magnetic properties.

At the end the authors state that they have found the “optimum annealing process”. That is not true, as they have only changed the annealing time and not the temperature. True is that they have obtained better samples that reported in literature obtained on similar prepared samples. Statement should be changed.

One point is not clear and should be addressed in more detail:

Although Sample B and C are prepared in same way (sample C is targeted check the reproducibility of results obtained by route II), they are different. Both consists of 2 phases: one rich in Mn (ε1) and one poor in Mn (ε2). Whereas sample B consists mainly of the Mn-rich phase (69%), Sample C consists mainly of the Mn-poor sample. On basis of this difference the difference in the obtained magnetic parameters is explained, but no comment is made, why Samples B and C are so different, although they are prepared in same way. It is not clear, how the separation in ε1 and ε2 was determined. In line 243 the authors state, that they have chosen the distribution and nominal composition of ε1 and ε2. Thus it seem not to be the result of a free fit in the Rietveld analysis. How these compositions were determined and what would be the result if other nominal compositions are chosen? Why the χ2-value for sample B is much higher than for sample C? these questions should be addressed, because the result is important for the interpretation of the magnetic hysteresis loops.

To help the reader and for better comparability of the results with those from literature the published ones should be given in the same units. (E.g. M is given in Am2/kg, kGs, kA/m, T)

The manuscript needs urgently a proof reading by a native English speaker.

There are also several typing errors. To mention a few:

Line 25: write “refinement”

Line 45: “magnets” is missing after “permanent”

Line 91: write “route, the” and “to optimize”

Line 97: wirte “Alloy ingots” and “refered”

Line 108 ff: “It sample of Mn …” is not a sentence. Meaning seems to be, that in first step Mn was molten, and in a second step molten Mn and compacted Al-C powder was molten. Sentences should be reformulated.

Line 149: write “Rietveld”

Line 152: write “through”

Line 185: add “Figure 1b” after Route II

Line 190: write “distribute”

Line 221: write “Rietveld”

Line 222: write “refinements”

Line 224: write “secondary”

Line 264: write “B and C” instead of “A and B”

Line 306: write “the” instead of “de”

Line 331: write “can be seen”

Line 352: write “Am2/kg”

Line 474: citation of ref. 44 is wrong!

Author Response

The Editor

Molecules Journal of the Multidisciplinary Digital Publishing Institute (MDPI)

November 7th, 2022

Dear Editor,

Thank you for considering our manuscript with ID: molecules-1984915 for publication in Molecules.  I am grateful to you and the reviewers for the valuable suggestions provided. They have all been considered and we have revised our manuscript following their suggestions.

Here are responses to the reviewer #2 comments:

Dear reviewers, we are very grateful for all your comments on our manuscript, which have led us to rethink some analyses, looking for the best presentation of the results that enable us to improve the quality of the manuscript. Major corrections were done, they were introduced in the manuscript using the “Track Changes” tool for Word.

  • At the end the authors state that they have found the “optimum annealing process”. That is not true, as they have only changed the annealing time and not the temperature. True is that they have obtained better samples that reported in literature obtained on similar prepared samples. Statement should be changed.

R// In the new version of the manuscript, we have removed this statement because, as the referee claims, we have not performed an extended and systematic study to find the optimum annealing process.

  • Although Sample B and C are prepared in same way (sample C is targeted check the reproducibility of results obtained by route II), they are different. Both consists of 2 phases: one rich in Mn (ε1) and one poor in Mn (ε2). Whereas sample B consists mainly of the Mn-rich phase (69%), Sample C consists mainly of the Mn-poor sample. On basis of this difference the difference in the obtained magnetic parameters is explained, but no comment is made, why Samples B and C are so different, although they are prepared in same way.

R// We believe that the difference in the concentration of Mn-rich and Mn-poor phases should affect the resulting crystalline microstructure after the e à t transformation, however we don’t have evidence of it (a detailed study is required). In the new version of the manuscript, we have attributed such differences to different resulting crystalline microstructure, in general, without discussing what cause such difference..

  • It is not clear, how the separation in ε1 and ε2 was determined. In line 243 the authors state, that they have chosen the distribution and nominal composition of ε1 and ε2. Thus it seem not to be the result of a free fit in the Rietveld analysis. How these compositions were determined and what would be the result if other nominal compositions are chosen?

R// The separation in the ε1 and ε2 phases was proposed based on several criteria: (1) According to the literature, local fluctuations on the Mn allows to induce a phase segregation of the e-phase with two different magnetic behaviors: Mn-rich regions which are antiferromagnetic, Mn-poor regions which are ferromagnetic. From a magnetic point of view, there must be at least 2 phases; (2) The nominal composition was chosen considering the upper and lower limits within the phase is stable, according to the Mn-Al phase diagram. The selection of these two limits would guarantee a Mn-rich and Mn-poor region; (3) The latter selection allowed a significant improvement of the refinement. The evaluation of multiple Mn-content combinations, and assume more than two e-phases, will require a dedicated and exhaustive study.

  • Why the χ2-value for sample B is much higher than for sample C? these questions should be addressed, because the result is important for the interpretation of the magnetic hysteresis loops.

R// We redid the refinement of t-Sample-B including possible secondary phases, but the refinement didn’t improved. We believe that it is necessary to include, at least, a second t-phase that contains most of the crystal defects which is assumed in this sample. Because we have no idea how this phase should look like, we prefer not to discuss it in this manuscript. 

  • To help the reader and for better comparability of the results with those from literature the published ones should be given in the same units. (E.g. M is given in Am2/kg, kGs, kA/m, T)

R// The different magnetization units were converted to the same units (Am2/kg), for which the theoretical density of the tau phase (5.04 g/cm3) was used.

  • The manuscript needs urgently a proof reading by a native English speaker. There are also several typing errors. To mention a few:

R// In this new version of the manuscript, that contains several changes, the english has been reviewed and corrected. In addition, all the references have been rechecked.

Round 2

Reviewer 1 Report

This new version of the manuscript addresses all the points I have underlined during the previous round of revision. The additional clarifications provided, in particular, the refined analysis of the Rietveld results and of the magnetic characterization of the samples reinforce the conclusions of this investigation, further improving the quality of this work. For this reason, I would recommend the publication of the manuscript in its present form.